# Vaccinated and Unvaccinated Risk Perceptions and Motivations for COVID-19 Preventive Measures Based on EPPM—A Polish Qualitative Pilot Study

**DOI:** 10.3390/ijerph192013473

**Published:** 2022-10-18

**Authors:** Katarzyna Domosławska-Żylińska, Magdalena Krysińska-Pisarek, Katarzyna Czabanowska, Giulia Sesa

**Affiliations:** 1Department of Education and Communication in Public Health, National Institute of Public Health NIH—National Research Institute, Chocimska 24, 00-791 Warsaw, Poland; 2Department of International Health, Care and Public Health Research Institute CAPHRI, Faculty of Health, Medicine and Life Sciences, Maastricht University, Duboisdomain 30, 6229 GT Maastricht, The Netherlands; 3Department of Health Policy Management, Institute of Public Health, Faculty of Health Sciences, Jagiellonian University, 31-066 Krakow, Poland

**Keywords:** COVID-19, risk perceptions, COVID-19 preventive measures, qualitative research, extended parallel process model, risk communication, Poland

## Abstract

COVID-19 has been a “major interrupting event” during which individuals largely relied on intuitive risk perceptions as precursors of COVID-19 health behaviors. Given the strong correlation between risk perceptions and the adoption of preventive measures, this pilot study attempts to explore how Polish society perceives COVID-19 health risks from the point of view of vaccinated and unvaccinated individuals. The Extended Parallel Process Model (EPPM) constitutes the basis for this research. Two focus group discussions (FGDs) were conducted, one with participants who had received COVID-19 vaccinations and the other with participants who had not. Conventional and directed content analyses were used to examine the data from the FGDs. Five categories emerged from the data: COVID-19 risk perceptions, coping with the pandemic, sources of knowledge, distrust, and pandemic fatigue. All categories fit into the theoretical constructs of the EPPM. Both groups have a high-perceived vulnerability to COVID-19 infection and recognize its seriousness. Individuals also have a high perceived response self-efficacy, given their awareness of COVID-19 preventive measures and how these should be applied. Nonetheless, particularly the unvaccinated, are skeptical about the effectiveness of the implemented measures, showing low perceived response efficacy. Future communication strategies should target the effectiveness of COVID-19 preventive measures, and one’s perceived response efficacy, to improve adherence to public health measures.

## 1. Introduction

The COVID-19 outbreak has been a “major interrupting event”, significantly affecting and changing how society and people deal with risk [1]. Specifically, the spread of contradictory, accurate, and inaccurate information, at an unprecedented speed and pace, led to a significant confusion among the public, to a lack of confidence towards official authorities, and to societal polarization [2,3]. Thus, individuals lacked guidelines on how to behave and majorly relied on intuitive risk evaluations, emphasizing the importance of risk perceptions [1]. For instance, despite extensive governmental recommendations, the Polish public has highlighted significant hesitation to follow governmental recommendations, such as vaccination. In terms of the percentage of fully vaccinated people in the EU, Poland is in 22nd place [4]. As of 30 June 2022, 22,511,049 individuals, or 58.8% of the Polish population, had completed the vaccination cycle; in other words, they received both vaccinations, or just one dose, of the Johnson & Johnson product [5]. 

Risk perceptions are subjective judgements that individuals make concerning the severity and characteristics of a risk. Such judgements are related to beliefs regarding the likelihood of loss or harm [6]. In other words, the term risk perception *“refers to people’s subjective judgments about the likelihood of negative occurrences such as injury, illness, disease, and death”* [7]. As individuals make decisions based on risk perceptions (rather than on effective risks), these are *“important precursors to health-related behaviors and other behaviors that experts recommend for either dealing with or preventing risks”* [7]. Specifically, Cori et al. (2020) [8] have established that compliance and non-compliance to COVID-19 guidelines are linked to one’s risk perceptions. Similarly, Abdelrahman (2020) [9] found that risk perceptions were predictors of social distancing and Nelson et al. (2020) [10] established a link between risk perceptions and self-quarantining. Health literacy has also been shown to be associated with the uptake of COVID-19 preventive measures [11]. However, this does not seem to be related to COVID-19 risk perceptions [12]. Therefore, given the strong link between risk perceptions and the uptake of preventive measures and its association with four key issues (demographic factors, individual factors, geographic factors and time), it is essential to investigate risk perceptions to conceptualize the COVID-19 spread [13]. From the available surveys conducted in Poland for the Polish society, the perception of the COVID-19 pandemic has more of a social than a health dimension [14]. In terms of mitigation behaviors, concerns about economic and health safety were associated with taking preventive measures. In contrast, male gender, lack of trust in online sources of information about COVID-19, and insufficient knowledge about the effectiveness of specific prevention measures were associated with negative attitudes of adherence to health behaviors [15]. 

In tandem with the study of risk perceptions in psychology, the discipline of risk communication evolved [7]. The latter refers to informing and persuading individuals on risks in a way that enables them to accurately perceive them and make decisions [7]. It is crucial to understand societal risk perceptions related to COVID-19 to design interventions to increase society’s compliance with public health measures; “*the public must accurately perceive that the COVID-19 is a serious health condition impacting their life in multiple ways and also must have a strong belief that one is personally susceptible to it at any time and in any locality*” [16]. 

Given the low vaccination uptake in Poland as well as in other Eastern European countries, it is crucial to investigate risk perceptions in the Polish context (IHME, 2021). A previous study by Sobkow and colleagues (2020) [17] found a relation between risk perceptions and intentions to conduct predictive actions related to COVID-19. To investigate risk perceptions in the Polish context, the present study employs the Extended Parallel Process Model (EPPM) (Witte, 1992) [18]. The choice of utilizing the EPPM as a foundation to this research derives from the work by Birhanu and colleagues (2021) who applied the EPPM to COVID-19. The EPPM is a health communication model building on the concepts of perceived response and self-efficacy, also called overall efficacy, and the subjective perception of susceptibility and severity, namely, the health threat [16]. Such concepts lead to message acceptance and desired behavior change. According to the model, messages promoted by risk communication campaigns shall include a “*mix of threat arousing messages*” [16] addressing the perceived seriousness of the consequences resulting from COVID-19 infection (PS) and perceived vulnerability to the infection (PV). Furthermore, public health messages shall target the ability of an individual to practice recommendations, namely the perceived response self-efficacy (PSE), and beliefs concerning the proposed solution’s effectiveness, namely the perceived response efficacy (PRE) [16]. Successively, when exposed to health messages, such as COVID-19 ones, an individual could be in danger control, exhibiting protective motivation responses, or fear control process, showing defending mechanisms from messages promoted by health campaigns [16]. A health risk (in this case COVID-19) can lead to either maladaptive/self-defeating or adaptive/self-proactive behaviors based on efficacy and threat levels [16]. From efficacy-threat interactions, four attitudinal groups result, namely, indifferent groups (low threat, low efficacy), avoidant (high threat, low efficacy), proactive (low threat, high efficacy), and responsive (high threat, high efficacy) [16]. Concerning COVID-19 those belonging to the responsive group favor the adoption of COVID-19 guidelines with a high motivation [16]. Those in the proactive category have low motivation levels, practicing minimal self-protective responses [16]. Contrastingly, the avoidant category presents defense mechanisms such as denial to COVID-19 guidelines and the indifferent group does not consider the issue of COVID-19 as relevant [16].

Finally, this pilot study attempts to explore how Polish society perceives COVID-19 health risks from the point of view of vaccinated and unvaccinated individuals. Findings from this pilot study can be a basis for designing larger-scale studies to improve knowledge of risk perceptions and motivations for preventive measures against COVID-19 and other infectious diseases.

## 2. Materials and Methods

To investigate COVID-19 risk perceptions among vaccinated and unvaccinated individuals in Poland, the researchers conducted a qualitative study, using the FGD (Focus Group Discussion) method. As defined by Morgan (1997) [19], this is a research technique that uses the cooperation and interaction of a group of participants to gather the necessary information. In this study, a classic focus group was used in terms of the number of participants, which was assumed to be 6–9 participants for Europe and 10–12 participants for the US [20]. (ref) In terms of duration, the interviews were extended beyond the standard hour due to the sensitivity of the topic and the EPPM-based scenario design, which required longer time to exhaust the topic. It was decided to use a qualitative method to explore behavioral insights and explore the decision-making process on COVID-19 issues.

### 2.1. Participant Recruitment

Purposive selection of participants was used, which is widely recommended, due to the participants’ ability and capacity to provide relevant information [21]. The recruitment process lasted between 29 November 2021 to 10 December 2021. Participants were recruited by an external company specializing in qualitative research. To locate possible participants, an external company contact to four general practice clinics from provinces with average vaccination rates (44–48% vaccinated people in the province; minimum 35%; maximum 52%; November 2021 [5]) were approached including, Lubusz, Lesser Poland, Silesia and West Pomerania regions. Participants applied to take part in the study based on information posted at clinics. In the next step, by telephone, a questionnaire (verifying the fulfillment of the participation conditions) was used to make the final verification and recruit participants. Each of the discussion groups assumed the participation of: a person who has undergone COVID-19, a person who has not undergone COVID-19, a person at risk (e.g., diabetes, age 65+, etc.), a working person with a family and children, a working person without a family (single), a young person 18–26 years old. The study’s target population included fully vaccinated and unvaccinated individuals living in cities with up to 250,000 inhabitants. Individuals who could not be immunized due to medical conditions were excluded. This selection of participants was aimed at building a creative group that would describe issues related to risk and motivation to deal with COVID-19 from different perspectives. 

### 2.2. Data Collection 

Two FGDs, consisting of vaccinated (*n* = 6) and unvaccinated (*n* = 6) individuals, were conducted. All interviews were conducted in Poland during the 4th wave of the COVID-19 pandemic (14 and 16 December 2021). Due to the epidemic situation, the interviews took place online, so that participants were in familiar surroundings, which provided a comfortable place for discussion and the ability to focus attention. The interviews were conducted by an external, professional moderator with over 10 years of experience in conducting this type of research. The FGDs lasted approximately 110 min (108 min vaccinated, 111 min unvaccinated). First, the researchers welcomed the participants and introduced the study. This was followed by an icebreaker game and an examination of participants’ expectations in relation to the FGD. During this stage, the researchers explained the importance of the study in bringing beneficial societal changes. Second, based on the EPPM three main discussion topics were formulated, such as, the perceived risk of COVID-19 infection, the enablers and obstacles to adhering to COVID-19 guidelines, and the risk communication needs of Polish citizens. Based on the three topics, a total of 12 open-ended questions were elaborated. Such questions were all constructed based on the EPPM (Table 1*).* Detailed questions were developed by researchers. The structure of the questions was to allow for an answer to the perceived seriousness of the consequences resulting from COVID-19 infection (PS), perceived vulnerability to the infection (PV), perceived response self-efficacy (PSE) and perceived response (PRE).

### 2.3. Data Analysis 

The data emerging from the FGDs were transcribed in Polish and then analyzed using conventional content analysis by three researchers separately (KDZ, KC and MKP). According to Hsieh and Shannon (2005) [22], conventional content analysis is an approach to data analysis where coding categories originate directly from interview transcripts. Table 2 summarizes the process of coding respondents’ statements via content analysis. Successively, directed content analysis, which, according to Hsieh and Sannon (2005) [22], entails using theory as guidance for initial coding, was used to match the obtained categories with the constructs from the theoretical model (i.e., the EPPM).

#### Quality Assurance 

The content of interviews was analyzed by three researchers to ensure the reliability of the findings and make sure that these are perceived as relevant. Finally, results were translated into English using back translation method to ensure that that the English transcripts adequately reflected the answers of the participants. 

## 3. Results

Five leading categories were obtained from the conducted content analysis: perceptions of COVID-19 risk (infection, complications and social consequences), coping with the pandemic, sources of knowledge, distrust, and pandemic fatigue. All categories fit into the EPPM.

### 3.1. COVID-19 Risk Perceptions

Both vaccinated and non-vaccinated persons perceive the risk of COVID-19 infection to be quite high. Immunized individuals point out that there is no way to prevent COVID-19 infection and that the risk of being infected is comparable to that of other infectious illnesses that are spread by droplets. 

Both groups agree that the virus is dangerous, posing a real threat to health. Those who have been vaccinated speak of threat in direct terms. Contrastingly, the unvaccinated gave examples of people with severe COVID-19 who are considered “specimens of health”.


*“The 30-year-old, who was athletic and had no chronic illnesses, was near death and was on a ventilator due to Covid”*
(Unvaccinated)

Both groups are aware of the health danger and socioeconomic issues that may arise resulting from COVID-19 and the pandemic. Particularly, the respondents emphasized the cardiovascular and neurological complications resulting from the infection, social functioning problems, and the educational gaps resulting from online learning, as illustrated in the below quotes. 


*“They often talk about the so called Covid fog which means that they have a harder time thinking, a harder time concentrating, and they also have a much high pulse rate and get tired faster”*
(Unvaccinated)


*“We begin to isolate ourselves from others and find it increasingly difficult to return to these contacts with others”*
(Vaccinated)


*“[COVID-19 leads to] irreversible gaps in education, in skills and social and knowledge”*
(Vaccinated)

Both groups drew attention to the role of immunity in the infection and the course of disease. As shown from the below quotes, the respondents believed that the risk of infection is an individual issue, with immunocompromised persons more susceptible to the disease. 


*“Infection depends on the individual because some people are more susceptible to this infection, others less and others not”.*
(Unvaccinated)


*“Complications depend on the strength of the body and maybe the experience of the disease and the years lived”.*
(Vaccinated)

### 3.2. Coping with the Pandemic

Both groups identified the following risk reduction activities to cope with the pandemic: wearing masks, washing and disinfecting hands, isolating the sick person as exemplified in the below quotes. 


*“I am vaccinated, and I wear a mask and I think others should do it too”.*
(Vaccinated)


*“The easiest in my opinion is disinfection”*
(Vaccinated)


*“Hand washing is good, and this is something we should continue”*
(Unvaccinated)


*“The most effective thing at the moment is not to leave the house sick”*
(Unvaccinated)

The groups surveyed believed that another element for coping with the pandemic is knowing how to manage the disease and how to cope if infected. In the case of contracting the illness, both groups state what would happen next, including; contacting a doctor, taking a test, undergoing isolation, and taking appropriate treatment.

Furthermore, accepting and learning to live with the virus is a way of coping with the COVID-19 pandemic. According to vaccinated individuals, immunizing against COVID-19 is a way to return to normality. Nonetheless, this will be a “new” normality (in other words, it will be different than before the pandemic). Opposingly, the unvaccinated suggested that returning to normality entails the cancellation of all restrictions and limitations. 


*“We have to learn to live with it [the virus] and this normality will be different than it was, but it will definitely come back to us”*
(Vaccinated)


*“So, let’s leave it alone let’s get back to life and not play with any restrictions”*
(Unvaccinated)

An adequate supply of information is needed to properly deal with the pandemic. Both groups noted the importance of having access to detailed statistics on incidence, deaths and vaccination rates. According to vaccinated individuals, presenting COVID-19 information in a true/false format would improve its readability. Moreover, they pointed out the importance of the availability of regional information and training for crisis information managers at the national level. To ensure the availability of accurate and timely information on COVID-19, a platform for sharing and exchanging disease experiences and educational campaigns would be a useful tool.

The study groups differ within their belief in the efficacy of COVID-19 prevention interventions and in how they manage the disease. For instance, the vaccinated, as opposed to the unvaccinated, believe that covid passports and vaccination are effective methods to reduce the risks associated with a pandemic. Contrastingly, the unvaccinated see it as a restriction of their civil liberties. 

The study groups also differ in their approach to treatment. Unvaccinated individuals reported a high willingness to self-treat with antipyretics and amantadine, despite their unproven efficacy for COVID-19 treatment:


*“If I have any suspicion, I take amantadine”*
(Unvaccinated)


*“I always take gripex”*
(Unvaccinated)

### 3.3. Sources of Knowledge

Both groups use similar sources of information for obtaining COVID-19 knowledge. However, the groups differ in the way they do so, as shown below. Firstly, both groups cited their own experiences and those of friends and family as sources of information about COVID-19 as shown below.


*“I passed the disease mildly”*
(Unvaccinated)


*“I had a sick person in my family and a friend but even though they have recovered they still have trouble carrying something, they get tired quickly”*
(Vaccinated)

Secondly, the radio and the Internet are other sources of knowledge. The unvaccinated consulted governmental information less than the vaccinated and were more likely to go to Twitter or Facebook for retrieving information.

Thirdly, both vaccinated and unvaccinated individuals stressed the importance of doctors in obtaining COVID-19 knowledge. Vaccinated people perceive the opinion of doctors as fundamental in taking decisions such as vaccinating. Conversely, unvaccinated people tend to contact a doctor when symptoms appear and there is no other way to deal with the disease. 


*“The doctor convinced me to get vaccinated”*
(Vaccinated)


*“Here there is no way out you have to report to a doctor”.*
(Unvaccinated)

Finally, as illustrated in the below quotes, in contrast to vaccinated persons, unvaccinated use information from unknown sources. They also used an anecdotal narrative in which the sender of the information was unknown. The lack of a sender or source of data reduces the ability to verify the cited examples and undermines their credibility. 


*“After the data as it is, after vaccination there is no 100% certainty that you will not get sick”.*



*“I was always told that if you were to get vaccinated you were to be healthy and now, they want to vaccinate everyone”.*


### 3.4. Lack of Trust

Both groups lack trust towards restrictions, information, government and authorities. According to both groups, some restrictions are incomprehensible, and their constant changes and the lack of clear interpretation create chaos. Vaccinated individuals noted that the introduction of stringent regulations is excessive. Additionally, the lack of verification of compliance with restrictions creates a sense of injustice.


*“and when I walk into a bank, for example, I meet a lady without a mask and I am required to wear one, and suddenly she is not required to wear one”.*
(Vaccinated)

Both groups of vaccinated and unvaccinated people lack trust in COVID-19 information, mainly due to its multiplicity, lack of consistency and transparency as shown below. 

“*I don’t believe the data that is put on the internet*” (Unvaccinated)

“*It is sometimes a pea with cabbage first there are one recommendation then another, fake news intermingle with real information*” (Vaccinated) 

In the statements of both groups, we can see a lack of trust in the government in terms of strategies to manage the pandemic and in COVID-19 regulations, given their low consistency. Unvaccinated individuals noted that the unpredictability of restrictions makes it hard to plan future tasks, as exemplified in the below quote. The two FGDs highlighted that the lack of trust towards government institutions is more evident among the unvaccinated.

“*You can’t plan anything, you don’t know if we’ll be shut down again in a while, or if they won’t introduce more restrictions*”.(Unvaccinated)

Compared to the vaccinated, the unvaccinated further lacked trust in physicians. According to them, doctors are no longer authorities as indicated by the quote below. The lack of medications to treat COVID-19 contributes to such trend (first quote). 

“*No pills, no medicine, no syrup nothing. As a sick person I wasn’t given any medication and I was just supposed to isolate myself. This is the only preventative measure for this disease which I think is absurd. Please find me a second, third and fifth disease that we have but we don’t treat it because it will pass by itself*” (Unvaccinated)

“*I think that today the greater authorities for many people will not be doctors but their favorite celebrities, actors, singers*” (Unvaccinated)

The unvaccinated compared to the vaccinated manifest a strong lack of confidence in vaccination in terms of efficacy and safety.

Among the factors contributing to such a lack of trust in vaccines are the little time employed in vaccine research, vaccination side effects (for instance, an increased risk of thrombotic disease), and an unknown overall safety profile. Moreover, the unvaccinated are discouraged from vaccination buses, which in their opinion look unprofessional (lack of proper examination before vaccination, feeling that the medical standard is lowered, lack of trust in the vaccination procedure in this form).

“*It used to be that in order to be vaccinated, one had to do a set of tests and there was a questionnaire for this and now there are vaccination buses and regardless of what health condition one has, one is vaccinated*”. (Unvaccinated)

“*I am not convinced that I can be vaccinated in a shopping mall or in a pharmacy, where no one is interested in me*”. (Unvaccinated)

### 3.5. Pandemic Fatigue 

Both study groups highlighted the emergence of pandemic fatigue in society, which is primarily manifested by a reduced perception of risk in relation to COVID-19 and a lower commitment to comply with sanitary restrictions:

“*People are tired and allow themselves to be a bit more relaxed*” (Vaccinated)

“*It got so after 2 years that people are fed up with it*” (Unvaccinated)

The categories obtained from the content analysis resemble the EPPM in all dimensions. Specifically, the category COVID-19 risk perceptions, which entails the risk of COVID-19 infection and the consequences resulting from it, reflects the model in the perceived vulnerability and perceived severity dimensions. Similarly, the category pandemic fatigue refers to the susceptibility of individuals to COVID-19 risks, matching the model in the perceived vulnerability dimension. Moreover, the categories coping with the pandemic and sources of knowledge refer to one’s knowledge of COVID-19 preventive measures corresponding to the perceived self-efficacy dimension. Lastly, the lack of trust category refers to one’s beliefs concerning the effectiveness of the implemented preventive measures, resembling the model in the perceived response efficacy dimension. A comparison of the obtained categories with the EPPM is displayed in Table 3. 

## 4. Discussion

Five key categories including perceptions of COVID-19 risk, coping with the pandemic, sources of knowledge, distrust, and pandemic fatigue emerged from the data. First, both vaccinated and unvaccinated individuals perceive the risk of COVID-19 infection as high and agree that contracting COVID-19 is dangerous. Second, both groups are fully aware of the various preventive measures in place and how these should be applied. Third, vaccinated and unvaccinated individuals retrieve COVID-19 knowledge from comparable sources. Nonetheless, such sources are used differently and are attributed a different value. Fourth, both groups are skeptical about the effectiveness of the restrictions applied throughout the pandemic, with the unvaccinated expressing more doubts. Fifth, the advent of pandemic fatigue reduces perceptions of COVID-19 as harmful and the chance of following preventive actions. 

The categories extrapolated from the content analysis and summarized in the above key findings connect to the EPPM. Specifically, this pilot research shows that both vaccinated and unvaccinated individuals have a high perceived vulnerability to COVID-19 infection, a high perceived seriousness of the consequences resulting from infection, a high perceived self-efficacy, and a low response efficacy. The latter applies particularly to the unvaccinated group. Such results demonstrate that individuals have low efficacy levels and high COVID-19 threat, being more prone to protect themselves from the fear rather than the actual risk. Therefore, individuals are in a fear control process. This finding is at odds with the studies by Jahangiry et al. (2020)] [23] and Vijeth et al. (2021) [24], which highlighted that individuals are mostly motivated by danger control, meaning they protect themselves from COVID-19 infection and adequately manage the risk. Other studies performed in the Nigerian and Ethiopian context, showed that the perceived threat surrounding COVID-19 is low. [25]. Conversely, this study detected a high perceived threat level. Moreover, this study found that individuals generally follow COVID-19 recommendations, despite mistrust surrounding them. Concerning the latter, this study found significant concerns surrounding vaccines’ safety and efficacy, especially among the unvaccinated, suggesting low response efficacy. Such a finding is at odds with the efficacy-threat interactions highlighted by the EPPM model. Particularly, with high threat and efficacy levels one would expect individuals to deny the COVID-19 threat and neglect guidelines. Nonetheless, according to the results of this study, individuals belong to the responsive and proactive attitudinal groups. Hence, the results of this study go against the initial propositions of the model. Such a result is not surprising given the argument that the propositions of the EPPM are not supported by sufficient empirical evidence (Popova, 2012) [26]. 

These findings seem to be unique as to our knowledge there are no studies investigating risk perceptions among the vaccinated and unvaccinated in the Polish context. This study improves the knowledge surrounding COVID-19 societal risk perceptions and the motivation of vaccinated and unvaccinated individuals to follow governmental recommendations, being particularly relevant in the Polish context. Specifically, the results of this research are pivotal in identifying future directions for communication strategies, increasing social responsiveness. The findings of this research can be used by the Polish government and the National Institute of Public Health to plan communication and information campaigns to reduce COVID-19’s burden. According to the findings of this pilot study, future health communication strategies shall extensively focus on delivering clear messages to the public, particularly emphasizing the effectiveness of the proposed measures. To do so, research results and scientific evidence for the implemented recommendations should be presented in a simple, easy-to-read form using statistical data, infographics, or true or false statements, exemplifying the effectiveness of interventions. Providing such evidence for the implemented recommendations can serve to increase trust towards COVID-19 guidelines, vaccines, authorities, and the government.

### 4.1. Strengths and Limitations

The small sample size of this study, explainable given the pilot nature of this work, does not ensure the generalizability of the findings to a Polish and broader European context. Nevertheless, to overcome such limitations, the researchers selected participants from various Polish regions. Another limitation of this research lies in the critiques surrounding the EPPM highlighted in the previous section. However, this study does not predict one’s behaviors towards health messages based on efficacy and threat levels, rather it also investigates whether individuals follow governmental guidelines making the relation between such dimensions more tangible.

This study also had several strengths, namely, its heavy reliance on the EPPM, which has been used for analogous studies, ensuring its relevance, participants’ diverse backgrounds, which ensured that different ideas and opinions were included in the study, and the parallel analysis of the data by three researchers’, which avoided subjectivity in data analysis.

### 4.2. Future Research 

Future research is recommended to explore COVID-19 risk perceptions among Polish regions with more that 250,000 inhabitants and include a higher number of interviewees per region. Similar studies could also be performed on the broader European context and explore how risk perceptions are correlated to education levels, age, socioeconomic status and other. 

## 5. Conclusions

This study demonstrated that in Poland vaccinated and unvaccinated individuals are concerned about contracting COVID-19 and its effects and generally follow COVID-19 recommendations. Nonetheless, both groups are skeptical about the effectiveness of the restrictions applied throughout the pandemic, with the unvaccinated expressing more doubts. Future communication strategies shall target more extensively the effectiveness of COVID-19 preventive measures, and hence one’s perceived response efficacy, to improve adherence to public health measures. 

## Figures and Tables

**Table 1 ijerph-19-13473-t001:** Research questions and specific questions used in the FGDs scenario.

Research Question	Detailed Questions
Based on current crisis communication, how do Polish citizens perceive the risks associated with the COVID-19 pandemic?	What is your opinion concerning the possibility of being infected with COVID-19? What is the chance of contracting COVID-19 despite being vaccinated according to you? How do you perceive COVID-19 concerning its seriousness and possible complications? Does COVID-19 have major adverse effects on one’s life? What sort, and why, (including: personal, professional, and financial)? What is your perception of the COVID-19 threat in the future (within the next 5 years)?
What are the crisis communication’s push and pull factors for adhering to the implemented COVID-19 recommendations?	Which recommendations do you think are effective in preventing Covid-19 infection? Are you able to follow the current recommendations and the advice of doctors/experts? Which recommendations and restrictions, in your opinion, were easy and which were difficult to follow, and why? What do you think about COVID-19 recommendations and restrictions in terms of transparency and legibility? In your opinion, was the information provided on the steps to be taken to reduce the risk of infection sufficient? In your opinion, were the recommendations unnecessary, e.g., too restrictive, and if so, why? If you need medical help with a suspected COVID-19 case, then…

**Table 2 ijerph-19-13473-t002:** The process of conventional analysis and coding respondents’ statement.

Level 1	Extracting categories of statements from the interview transcript of vaccinated and unvaccinated individuals
	Risk perception; Coping with COVID-19; Vaccination; Knowledge sources; Lack of trust; Pandemic fatigue
Level 2	Content analysis of the interviews in terms of the categories adopted separately for vaccinated and unvaccinated individuals and identification of subcategories
	Risk perception: the possibility of contracting, health complications and possibility of socioeconomic consequences of COVID-19 pandemic Coping with COVID-19: mask, disinfection, distance, isolation; covid passports, vaccination; disease management acceptance and learning to live with COVID-19, supply of information Vaccinations: safety, effectiveness Knowledge sources: own experiences, family and friends, Internet, radio, physicians, unconfirmed source Lack of trust: restrictions, information, government and authorities, physicians, COVID-19 treatment, vaccinations Pandemic fatigue
Level 3	Identify similarities and differences in the statements of vaccinated and unvaccinated individuals
Level 4	Synthesis of the vaccinated and unvaccinated statements and conversion of results in relation to the EPPM model
	Risk perception (PS and PV) Coping with COVID-19 (PRE) Sources of knowledge (PRE) Lack of trust (PSE) Pandemic fatigue (PV)

**Table 3 ijerph-19-13473-t003:** The crosswalk between emergent categories and constructs the EPPM by vaccinated and unvaccinated individuals.

Categories from FGDs	EPPM	Vaccinated	Unvaccinated
Perceptions of COVID-19 risks (infection, complications and social consequences)	Perceived seriousness (PS)	Awareness of the risk of consequences Role of immunity	
	Perceived vulnerability (PV)	Real high risk of contracting Role of immunity	
Pandemic fatigue		Getting used to a virus Tiredness of following the restrictions	
Coping with the pandemic	Perceived self-efficacy (PSE)	Mask, disinfection, distance, isolation Acceptance and learning to live with COVID-19 Supply of information	
		COVID-19 passports Vaccination Disease management with physician	Willingness to self-treat
Sources of knowledge		Own experiences Family and friends Internet Radio Physicians	
			Unconfirmed source
Lack of trust	Perceived response efficacy (PRE)	Restrictions, information, government and authorities	
			Physicians Treatment of COVID-19 Vaccinations

## Data Availability

The data analyzed during the current study are available.

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
