# Peer review of "Vaccinated and Unvaccinated Risk Perceptions and Motivations for COVID-19 Preventive Measures Based on EPPM—A Polish Qualitative Pilot Study"

_ijerph, 2022, doi:10.3390/ijerph192013473_

Round 1

Reviewer 1 Report

This is a very interesting qualitative study exploring how COVID-19 risk  is perceived by COVID-19 vaccinated and unvaccinated people. The article is well written and only minor revisions are required.

Introduction

Pg 2 ll 48 – 54. The role of health literacy as an important precursors of the adherence to COVID-19 preventive measures and its relationship with COVID-19 risk perception should be acknowledged. Indeed, health literacy has been shown to be associated to the uptake of COVID-19 prevention measures as these measures mostly rely on the ability of individuals to understand and adopt the correct preventive behaviors and to respect the rapidly evolving public health measures; interestingly, the health literacy level seems to be not related to the level of COVID-19 risk perception.

Suggested references:

- Li X, Liu Q. Social Media Use, eHealth Literacy, Disease Knowledge, and Preventive Behaviors in the COVID-19 Pandemic: Cross-Sectional Study on Chinese Netizens. J Med Internet Res. 2020 Oct 9;22(10):e19684. doi: 10.2196/19684.

- Lastrucci V, Lorini C, Del Riccio M, Gori E, Chiesi F, Moscadelli A, Zanella B, Boccalini S, Bechini A, Puggelli F, Berti R, Bonanni P, Bonaccorsi G. The Role of Health Literacy in COVID-19 Preventive Behaviors and Infection Risk Perception: Evidence from a Population-Based Sample of Essential Frontline Workers during the Lockdown in the Province of Prato (Tuscany, Italy). Int J Environ Res Public Health. 2021 Dec 19;18(24):13386. doi: 10.3390/ijerph182413386.

Pg 2 ll 68. Please include the reference to the original article in which the “Extended Parallel Processing Model”  was first formulated.

Methods

PG 3. Ll98-104 Please provide further details concerning the recruitment of participants.

Pg 3 ll 106. FGDs acronym should be expanded as this is the first time that is mentioned in the text.

Author Response

Response to Reviewer 1 Comments

Point 1: Pg 2 ll 48 – 54. The role of health literacy as an important precursors of the adherence to COVID-19 preventive measures and its relationship with COVID-19 risk perception should be acknowledged. Indeed, health literacy has been shown to be associated to the uptake of COVID-19 prevention measures as these measures mostly rely on the ability of individuals to understand and adopt the correct preventive behaviors and to respect the rapidly evolving public health measures; interestingly, the health literacy level seems to be not related to the level of COVID-19 risk perception.

Response 1: Thank you for this remark. It is indeed very important to acknowledge the role of health literacy and its connection to risk perceptions. The sentence ¨ Health literacy has also been shown to be associated to the uptake of COVID-19 preventive measures (Li & Liu, 2020). However, this does not seem to be related to COVID-19 risk perceptions (Lastrucci et al., 2021)¨ has been added to the text (pg 2 ll 54 – 56).

Point 2: Pg 2 ll 68. Please include the reference to the original article in which the “Extended Parallel Processing Model”  was first formulated.

Response 2: Thank you for this remark. The author of the EPPM has been added to the indicated sentence as follows: ¨To investigate risk perceptions in the Polish context, the present study employs the Extended Parallel Process Model (EPPM) (Witte, 1992)¨ (p. 2 ll 70).

Point 3: PG 3. Ll98-104 Please provide further details concerning the recruitment of participants.

Response 3: Thank you for this valuable remark. The section on recruiting participants has been supplemented with the dates of recruitment and detailed information on the organization of recruitment has been added. Minor changes to the order in the text have been made to increase the clarity of the sections

Point 4: Pg 3 ll 106. FGDs acronym should be expanded as this is the first time that is mentioned in the text.

Response 4: Thank you for this valuable remark. The acronym FGD has been expanded at the indicated place

Reviewer 2 Report

The manuscript is on the whole well written. However, there are some fixes to make:

What does "FGDs" mean? The authors should explain this better. Furthermore, the abbreviations should be repeated in full also in the introduction as well as in the abstract.

Table 1, were the questions developed based on guidelines? What is the criterion for choosing the questions? The authors should explain this.

The article as a whole is well written and structured, the results are clear. However, in discussion the authors should clarify why there is hesitation in the vaccination campaign. Definitely for the potential side effects, so they will have to mention these studies:

https://doi.org/10.1016/j.mayocp.2020.12.024

https://doi.org/10.3390/vaccines9030300

10.3324 / hematol.2021.279075

10.3390 / diagnostics11060955

https://doi.org/10.1016/j.puhe.2021.02.025

Author Response

Response to Reviewer 2 Comments

Point 1: What does "FGDs" mean? The authors should explain this better. Furthermore, the abbreviations should be repeated in full also in the introduction as well as in the abstract.

Response 1: Thank you for this valuable remark. In the material and methods section, an explanation of the focus group discussion technique has been added. The acronym FGD has been expanded at the indicated place.

Point 2: Table 1, were the questions developed based on guidelines? What is the criterion for choosing the questions? The authors should explain this.

Response 2: Thank you for this valuable remark. We have included in the manuscript that detailed questions were developed by researchers. The structure of the questions was to allow for an answer to the perceived seriousness of the consequences resulting from COVID-19 infection (PS), perceived vulnerability to the infection (PV), perceived response self-efficacy (PSE) and perceived response (PRE).

Point 3: The article as a whole is well written and structured, the results are clear. However, in discussion the authors should clarify why there is hesitation in the vaccination campaign. Definitely for the potential side effects, so they will have to mention these studies:

https://doi.org/10.1016/j.mayocp.2020.12.024; https://doi.org/10.3390/vaccines9030300; 10.3324 / hematol.2021.279075; 10.3390 / diagnostics11060955; https://doi.org/10.1016/j.puhe.2021.02.025

Response 3: Thank you for this valuable remark. In response to comments from other reviewers, we removed the vaccination hesitancy fragment due to the fact that it is not the main goal of the study, focusing in the discussion on the reference to the EPPM model..

Reviewer 3 Report

Intruducion

There are no refererences to research studies on attitutes towards 
Covid-19, vaccination in Poland and other countries. 

EPPM is not analysed in detail and it has some limitations and sometimes is criticized - compare Popova L (August 2012). "The extended parallel process model: illuminating the gaps in research". Health Education & Behavior39 (4): 455–473..

There is missing discusion of model

Missing information on qualitative research methodology approach selection

Methods

Two focus groups seems to not enough for this kind of research studies, even it is pilot study; saturation approach should be taken into consideration in small number of groups (please check: Guba G.E. (1981), Criteria for Assessing the Trustworthiness of Naturalistic. Educational Communication and Technology, 29 (2), 75-91. Guest G., Bunce A. & Johnson L. (2006), How Many Interviews Are Enough? An Experiment with Data Saturation and Variability, Field Methods, 18 (1), 59-82. Marshall B., Cardon P., Poddar A., Fontenot R. (2013). Does sample size matter in qualitative research?: A review of qualitative interviews in is research, Journal of Computer Information System, 54 (1), 11-22. Mason M.(2010), Sample Size and Saturation in PhD Studies Using Qualitative. Forum Qualitative Sozialforschung, 11 (3) / Forum: Qualitative Social Research, Art. 8, http://nbn-resolving. de/urn:nbn:de:0114-fqs100387

Usually 2 groups per homogenous group is regarded min in qualitative research methodology

 Missing information about methodology: traditional or online focus groups, who has moderated groups, what is the experience in moderating groups (which is crucial in qualitative research); precise timing of research; explanation for sample selection

Results:

two much citations; limited analyses - probably due to limited reseach material; 

Conlclutions: 

Very limited

Author Response

Response to Reviewer 3 Comments

Point 1: There are no refererences to research studies on attitutes towards Covid-19, vaccination in Poland and other countries.

Response 1: Thank you for this suggestion. We did not include Polish studies regarding vaccination hesitancy due to the fact that it is not the main purpose of our article (we only point out the fact of vaccination hesitancy) - this could cause unnecessary confusion. There is a very limited amount of research on COVID-19 risk perception in Poland. We added the revelant studies we found to the article. We've also added information on COVID-19 risk perceptions in other countries, by adding relevant references.

Point 2: EPPM is not analysed in detail and it has some limitations and sometimes is criticized - compare Popova L (August 2012). "The extended parallel process model: illuminating the gaps in research". Health Education & Behavior. 39 (4): 455–473.

Response 2: Thank you for this valuable remark and the article provided. Some details concerning the model have been added in the introduction. Based on the article by Popova (2011) (in particular, based on Table 2), the four attitudal groups resulting from efficacy-threat interactions have been better described by indicating the threat and efficy level corresponding to each (p. 2 ll 85 – 87). All the information and relationships highlighted in the introduction was checked based on image 1 provided by Popova (2011). Furthermore, to include the limitations emphasized by Popova (2011) the discussion as well as the limitations of our paper (p. 9 ll 352 – 361 and p. 10 ll 380 – 384).

Point 3: There is missing discusion of model

Response 3: Thank you for this remark. To include a discussion, our findings were compared to the EPPM (p. 9 ll 352 – 361).

Point 4: Missing information on qualitative research methodology approach selection

Response 4: Thank you for this valuable remark. We have supplemented the text with detailed information regarding the choice of FGD method

Point 5: Two focus groups seems to not enough for this kind of research studies, even it is pilot study; saturation approach should be taken into consideration in small number of groups (please check: Guba G.E. (1981), Criteria for Assessing the Trustworthiness of Naturalistic. Educational Communication and Technology, 29 (2), 75-91. Guest G., Bunce A. & Johnson L. (2006), How Many Interviews Are Enough? An Experiment with Data Saturation and Variability, Field Methods, 18 (1), 59-82. Marshall B., Cardon P., Poddar A., Fontenot R. (2013). Does sample size matter in qualitative research?: A review of qualitative interviews in is research, Journal of Computer Information System, 54 (1), 11-22. Mason M.(2010), Sample Size and Saturation in PhD Studies Using Qualitative. Forum Qualitative Sozialforschung, 11 (3) / Forum: Qualitative Social Research, Art. 8, http://nbn-resolving. de/urn:nbn:de:0114-fqs100387

Usually 2 groups per homogenous group is regarded min in qualitative research methodology

Response 5: Thank you for this valuable remark. The study, on the basis of which we decided on this number of interviews, shows that achieving code saturation captures a wide range of issues and requires a small number of focus groups. The majority of new codes (60%) were identified in the first focus group discussion. As the researchers themselves point out, one focus group per stratum was needed to identify issues. In the case of the pilot study, given the above findings, in order to identify the most important codes (issues) a single group is sufficient. Hennink MM, Kaiser BN, Weber MB. What Influences Saturation? Estimating Sample Sizes in Focus Group Research. Qual Health Res. 2019 Aug;29(10):1483-1496. doi: 10.1177/1049732318821692.

Point 6: Missing information about methodology: traditional or online focus groups, who has moderated groups, what is the experience in moderating groups (which is crucial in qualitative research); precise timing of research; explanation for sample selection

Response 6: Thank you for this remark. The above issues have been completed in the participants recruitment and data collection section.

Point 7:  Results: two much citations; limited analyses - probably due to limited reseach material;

Response 7: Thank you for this remark. In the text, we have removed some of the quotes, but due to the fact that this is a qualitative study, based on the statements of the participants, the quotes are the results of the research. We cannot omit them in order to show direct content from the interviews, as we will lose context and source data.

Point 8:  Conlclutions: Very limited

Response 8: Thank you for this suggestion. The conclusions are the essence of the study. They are written in a compact and concrete manner. Considering the very limited amount of research on Polish society in the context of COVID-19 risk perception and social behavior in the face of pandemic restrictions, our study is a significant contribution to the development of knowledge in the field of behavioral insights. The choice of a qualitative method allowed us to explore the topic at hand. Through the pilot study, we have outlined the possibilities for further research on this topic. We hope that the changes made to the manuscript will enhance the value of the article.

Round 2

Reviewer 2 Report

The article has been sufficiently improved. However, there are minor revision. Line 54, 55 it is useless to put the words (Li & Liu, 2020) and (Lastrucci et al., 2021), it is enough to put only the reference, according to the template.

Author Response

Remark: Line 54, 55 it is useless to put the words (Li & Liu, 2020) and (Lastrucci et al., 2021), it is enough to put only the reference, according to the template. 

Response: Thank you for this remark. We have removed the in-text references. 

Reviewer 3 Report

I advice in future to complete more groups as 2 is not really enough.

Author Response

Remark: I advice in future to complete more groups as 2 is not really enough.

Response: Thank you for this suggestion. We will keep it in mind for future studies.